



# Estimating the size of a methane emission point-source at different scales: from local to landscape

Stuart N. Riddick[1,*], Sarah Connors[1], Andrew D. Robinson[1], Alistair J. Manning[2], Pippa S. D. Jones[1], David Lowry[3], Euan Nisbet[3], Robert L. Skelton[4], Grant Allen[5], Joseph Pitt[5] and Neil R. P. Harris[6]

[1] Centre for Atmospheric Science, University of Cambridge, Cambridge, CB2 1EZ, UK
[2] Met Office, Exeter, EX1 3PB, UK
[3] Department of Earth Sciences, Royal Holloway, University of London, Egham, TW20 0EX, UK
[4] Department of Chemical Engineering, University of Cambridge, Cambridge CB2 3RA, UK
[5] Centre for Atmospheric Science, University of Manchester, Manchester, M13 9PL, UK
[6] Centre for Atmospheric Informatics and Emissions Technology, Cranfield University, Cranfield, MK43 0AL, UK
* Now at Department of Civil and Environmental Engineering, Princeton University, Princeton, 08544, USA

*Correspondence to*: Stuart N. Riddick (sriddick@princeton.edu)

**Abstract.** High methane ($CH_4$) mixing ratios (up to 4 ppm) have occurred sporadically at our measurement site in
Haddenham, Cambridgeshire since July 2012. Isotopic measurements and back trajectories show that the source is the
Waterbeach Waste management park 7 km SE of Haddenham. To investigate this further, measurements were made on June
30th and July 1st 2015 at other locations nearer to the source. Landfill emissions have been estimated using three different
approaches (WindTrax, Gaussian plume, and NAME InTEM inversion) applied to the measurements made close to source
and at Haddenham. The emission estimates derived using the WindTrax and Gaussian plume approaches agree well for the
period of intense observations. Applying the Gaussian plume approach to all periods of elevated measurements seen at
Haddenham produces year-round and monthly landfill emission estimates. The estimated annual emissions vary between
11.6 and 13.7 Gg $CH_4$ yr$^{-1}$. The monthly emission estimates are highest in winter (2160 kg hr$^{-1}$ in February) and lowest in
summer (620 kg hr$^{-1}$ in July). These data identify the effects of environmental conditions on landfill $CH_4$ production and
highlight the importance of year-round measurement to capture seasonal variability in $CH_4$ emission. We suggest the
landscape inverse modelling approach described in this paper is in good agreement with more labour-intensive near-source
approaches and can be used to identify point-sources within an emission landscape to provide high-quality emission
estimates.

## 1 Introduction

Atmospheric methane ($CH_4$) gas is both a greenhouse gas and partially responsible for modulating tropospheric ozone
production and loss. As such, changes in atmospheric $CH_4$ mixing ratios can cause significant shifts in local and regional
atmospheric chemistry and global climate. Current research suggests the most significant $CH_4$ sources are natural wetlands
(top-down, 142–208 Tg $CH_4$ yr$^{-1}$; and bottom-up, 177–284 Tg $CH_4$ yr$^{-1}$) and agriculture and waste emissions (top-down,





180–241 Tg $CH_4$ yr$^{-1}$; and bottom-up, 187–224 Tg $CH_4$ yr$^{-1}$), with further contributions from fugitive emission due to the use of fossil fuels, natural emissions and biomass burning (IPCC., 2013; Kirschke et al., 2013). Anthropogenic sources contribute ~60% of modern-day emissions (Saunois et al. 2016). Included in these estimates, decomposition of organic matter at landfills is estimated to comprise between 3% and 19% of global anthropogenic emissions (Chen & Prinn, 2006).

Given this large and important uncertainty, the aim of this study is to estimate $CH_4$ mass flux from an operational landfill in Cambridgeshire using a variety of methods.

Approximately 60% of gas emitted from typical landfills is $CH_4$, 40% is carbon dioxide and trace amounts are given off as nitrogen, oxygen and water vapour (Hegde et al., 2003). At the surface, anoxic microbial processes form $CH_4$, whereas oxidation forms both carbon dioxide and water. Deeper below the surface anaerobic processes dictate gas formation due to

the oxygen-poor environment. Simple organic acids (e.g. carboxylic acid), carbon dioxide ($CO_2$) and hydrogen ($H_2$) are formed from the hydrolysis of organic matter. Methanogenic bacteria then convert carboxylic acid (RCOOH) to $CH_4$ which can diffuse through the refuse to be emitted to the atmosphere (Xu et al., 2012). Riddick et al. (2016) suggest that instead of heterogeneous emission across the landscape landfill, $CH_4$ is emitted in discrete hot-spots which may be caused by variability in the materials that can degrade to form $CH_4$ throughout the landfill and the nature of physical transmission

pathways to the surface. Modern landfills in the UK have extensive reticulations of gas pipes to extract methane, and fractures or leaks in the pipes create potent point sources of methane to escape past the soil oxidation barrier.

The emitted $CH_4$ can be identified by measuring its $\delta^{13}C$ isotopic signature. Typically, biogenic methane has a $\delta^{13}C$ isotopic signature of between -55 and -70‰ (Dlugokencky et al., 2011). However, landfill methane emissions, which comprise the residual gas after the methane flux has passed through the oxidation barrier in the soil cover, tend to fall at the isotopically

heavier end of this range as oxidative methanotrophy is selective for the lighter carbon. Typically, the $\delta^{13}C$ isotopic signature for landfill $CH_4$ in the south east of the UK has been measured at -58 ± 3‰ (Zazzeri et al., 2015).

Although landfill interiors are well-isolated from day-to-day weather, and even seasonality, emissions from the landfill surface can be strongly affected by environmental conditions. Xu et al. (2012) and Riddick et al. (2016) observed decreasing landfill $CH_4$ emission as surface atmospheric pressure increased at landfill sites in Lincoln, USA and Ipswich, UK,

respectively. Emission of landfill $CH_4$ may be suppressed as atmospheric pressure increases; conversely, the passage of depressions may pneumatically draw gas out from the landfill. Landfill $CH_4$ emissions decrease with increased ground temperature in dry soil conditions (Scheutz et al., 2004; Riddick et al., 2016). This is consistent with the hypothesis that bacterial methanotrophic oxidation of methane in the aerobic cover soil has an Arrhenius relationship with temperature, increasing exponentially with ground temperature between 2 and 25 °C (Maurice & Lagerkvist, 2004; Scheutz et al., 2004).

A variety of methods have been used to estimate $CH_4$ emission estimates from landfill sites using on-site and near-site measurements. These include chamber methods, tracer plume and eddy covariance. Tracer release (TR) methods have been used to good effect, where pollutant mixing ratios are estimated using the co-release of a tracer at a known rate. However, this methodology needs the spatial distribution of tracer emissions to be configured so that it approximately matches the landfill $CH_4$ emissions (Mønster et al., 2014), presenting logistical challenges when operating on active landfill sites.





Landfill $CH_4$ emissions have been measured using eddy covariance techniques, which use the covariance between vertical wind speed and gas mixing ratio to estimate emissions at a high sampling rate (Xu et al., 2012). However, the assumption of homogeneity by eddy covariance calculations is invalidated by the heterogeneous nature of landfill $CH_4$ emissions. Furthermore, these estimates strictly apply to the area and time where the measurements are made. Estimates produced in a

heterogeneous environment such as a landfill can thus be hard to interpret or extrapolate to the whole landfill and to other times of year.

In this study we use methane measurements made at Haddenham, Cambridgeshire in which we record intermittently high values of up to 4 ppm when the wind is from the southeast. Methane emissions from the Waterbeach Landfill site, 7 km to the SE of our measurement site at Haddenham, are a likely source of these enhancements. To aid identification of this $CH_4$

source, we collected air samples during a south-easterly air flow and measured the relative abundance of $\delta^{13}C$ isotopes. These are compared with additional measurements made nearer the landfill. Short time series of $CH_4$ measurements taken near the landfill are used to estimate emissions using the inverse dispersion model WindTrax (www.thunderbeachscientific.com). The emissions are compared with a Gaussian plume estimate made using the Haddenham data for the same period. The Gaussian plume calculations are extended to cover the whole of the first two

years of measurements at Haddenham in order to investigate how the emissions vary over time. Finally, we aim to compare the annual emission estimate found using the Gaussian model with the estimate from the NAME InTEM inversion model that uses two years' $CH_4$ measurement data from a network throughout East Anglia to estimate the regional annual emission. The measurement and modelling techniques used are described in Sect. 2. The modelling studies performed are described in Sect. 3. The results are then presented in Sect. 4. The paper concludes with a short discussion and the conclusions of the

results and the broader applicability of the approach.

**2 Methods**

This presents methane emission estimates from a landfill made by three methods at different scales: near-source, middle-distance and landscape, a summary of each method is presented in Table 1. Waterbeach Waste Management Park (52.302 N, 0.180 E) is used to deposit unrecyclable waste on an open active area approximately 700 m by 300 m. Surrounding the

active area is an area of decomposing waste capped with a welded high-density polyethylene (HDPE) geo-membrane and covered with at least two meters of top soil. Landfill gas is extracted from this capped area under suction using a network of pipes and wells and is used as fuel for the on-site electricity generators. The various measurement techniques are now described in turn.

**2.1 Isotopic methane measurements**

Whole air samples were collected in 3L Teflon bags at Haddenham Church (Fig. 1). These samples were taken over the 11[th] February 2015 when the wind was from the south/south-east, i.e. from the direction of the landfill. Air samples were taken at



Haddenham in the early morning in order to capture the elevated mixing ratio of landfill emissions within the nocturnal boundary layer. The carbon isotopic ratio, expressed in ‰, was measured in triplicate to high precision (±0.05‰) by continuous flow gas chromatography isotope ratio mass spectrometry (CF GC-IRMS) (Fisher et al., 2006), at Royal Holloway, University of London (RHUL).

## 2.2 Near-Source

### 2.2.1 Measurements – Los Gatos UGGA

The Los Gatos Research Ultra-portable Greenhouse Gas Analyser (UGGA; www.lgrinc.com) is a laser absorption spectrometer that measures $CH_4$ and $CO_2$ concentration in air using off-axis integrated cavity output spectroscopy (Paul et al., 2001). The UGGA reports $CO_2$ mixing ratio and $CH_4$ mixing ratio every second, with a stated precision of < 2 ppb (1σ @ 1 Hz) over an operating range of 0.1 to 100 ppm. Calibration of the UGGA was done before and after deployment using low (1.93 ppm), target (2.03 ppm) and high (2.74 ppm) gases calibrated on the WMO scale.

The UGGA was deployed on a farm road on Mitchell Hill Farm, Cottenham (52.304 N, 0.170 E) where it measured the mixing ratio of $CH_4$ downwind of the landfill. The measurement site was 300 m NW of the landfill site. The inlet line was attached to a mast 2.5 m above the ground, protected from water incursion using an aluminium funnel and filtered using a 2 µm filter.

### 2.2.2 Meteorological Data

In situ meteorological data were collected using a wireless weather station (Maplin, UK) attached to a mast at 2 m from the ground at the measurement site on Mitchell Hill Farm. Meteorological data were sampled and recorded at five-minute intervals and include: wind speed ($u$, m s$^{-1}$), wind direction ($WD$, ° to North), air temperature at 2 m ($T_a$, K), relative humidity ($RH$, %), rain rate ($R$, mm hr$^{-1}$) and air pressure ($P$, Pa).

Micrometeorological parameters used for subsequent modelling were calculated from data collected at the same measurement site on Mitchell Hill Farm. Roughness height ($z_0$, m) and Monin-Obukhov length ($L$, m) are calculated from the wind speeds measured at three heights. The roughness length is calculated as the exponential of the intercept, with the natural logarithm of wind measurement heights plotted against wind speeds. The Monin-Obukhov length is calculated (Eq. 1) from the density of air ($\rho$, kg m$^{-3}$), the specific heat capacity of air at constant pressure ($c_p$, J kg$^{-1}$ K$^{-1}$), the absolute temperature of air at z = 0 ($T_0$, K), the acceleration due to gravity ($g$, m s$^{-1}$), and the sensible heat flux ($H$, W m$^{-2}$). The sensible heat flux ($H$, W m$^{-2}$) is calculated (Eq. 2) from the transfer coefficient for heat flux ($CH$, 1x10$^{-3}$) (Pan et al., 2003).

$$L = -\frac{\rho c_p T_0 u_*^3}{K g H} \quad (1)$$

$$H = \rho c_p CH (T_a - T_0) u \quad (2)$$





### 2.2.3 Model used – WindTrax Inverse Dispersion Model

The inversion function of the WindTrax atmospheric dispersion model version 2.0 (Flesch et al., 1995) is used to infer the $CH_4$ emissions from the landfill. Methane emissions are calculated using measured $CH_4$ mixing ratio enhancement downwind, measured background $CH_4$ mixing ratios upwind and the simulated ratio of $CH_4$ mixing ratio enhancement to

emission (Flesch et al., 2004; 2005). WindTrax calculates the ratio of $CH_4$ mixing ratio to emission by back-calculating the movement of many $CH_4$ particles from the detector to the landfill emission area and estimating the vertical velocity as they leave the emission area. Following the method of Laubach et al. (2008) and Flesch et al. (2009), $CH_4$ mixing ratios and meteorological data were averaged over 15 minutes to preserve real changes to $CH_4$ emission caused by changing environmental or atmospheric factors. Each 15-minute-averaged measurement is used as an input to back-calculate the $CH_4$

emission using 50,000 particle trajectories.

### 2.3 Middle-Distance

### 2.3.1 Measurements – GC-FID

Methane mixing ratios were measured every 75 seconds from July 2012 to July 2015 at the Holy Trinity church, Haddenham (52.359° N, 0.148° E) since July 2012 (see Fig. 1) using a 200 series Ellutia GC-FID (www.ellutia.com). The site elevation

is 40 metres above sea level and the inlet is on the tower, 25 m above the ground. The GC-FID takes air to be assayed for $CH_4$ mixing ratio mixed with a carrier gas which passes through a column of alumina coated tubing heated in an oven at 90°C. As the gases exit the column they are pyrolyzed by a hydrogen/air mixture within the flame ionization detector. Ions formed during the combustion are measured to indicate the mixing ratio of the gas species. The Ellutia GC-FID, as used here, has a detection limit of approximately 1.5 ppb, a range of 1.5 to 3 ppm and measures mixing ratios every 75 s. The

instrument is calibrated every 30 minutes using a gas standard. The Teflon inlet line is attached to the church roof 30 m above the ground and is protected from water incursion using an aluminium funnel and a 2 µm particle filter.
The data are transmitted data back to the laboratory for processing. Data processing of individual chromatograms is done using IGOR Pro (Wavemetrics, USA) to determine peak height. Measurements from all sites are calibrated to the WMO (World Meteorological Office) calibration scale (Dlugokencky et al., 2005). Hourly WMO calibrated mixing ratios are then

calculated using Openair in R.

### 2.3.2 Meteorological Data

Data were taken from UK Met Office's Numerical Atmospheric Modelling Environment (NAME) model, as described later in Sect. 2.4.2.





### 2.3.3 Model used – Gaussian Plume

The Gaussian Plume (GP) model describes the mixing ratio of a gas as a function of distance downwind from a point source (Seinfeld and Pandis, 2006). As a gas is emitted, it is entrained in the prevailing ambient air flow and disperses in the y and z directions (relative to a mean horizontal flow in the x direction) with time, forming a cone. The gas is considered to be well

mixed within the volume of the cone, such that the mixing ratio of the gas as a function of distance downwind depends on the emission flux at source, the advective wind speed ($u$, m s$^{-1}$), and the rate of dispersion (governed by boundary layer micrometeorological factors described in Sect. 2.2). The mixing ratio of the gas ($X$, μg m$^{-3}$), at any point x metres downwind of the source, y metres laterally from the centre line of the plume, and z metres above ground level can be calculated (Eq. 3) using the source strength ($Q$, g s$^{-1}$), the height of the source ($h_s$, m) and the air stability. The standard deviation of the lateral

($\sigma_y$, m) and vertical ($\sigma_z$, m) mixing ratio distribution are calculated from the stability class of the air (Pasquill, 1974). The Gaussian plume approach assumes that the vertical eddy diffusivity and wind speed are constant and there is total reflection of methane at the surface (e.g. Zannetti, 1990; Hensen and Scharff, 2001; Hensen et al., 2009).

$$Concentration\ (x, y, z) = \frac{Q}{2\pi u \sigma_y \sigma_z} e^{-\frac{y^2}{(2\sigma_y)^2}} \left( e^{-\frac{(z-h_S)^2}{(2\sigma_z)^2}} + e^{-\frac{(z+h_S)^2}{(2\sigma_z)^2}} \right) \quad (3)$$

### 2.4 Landscape

### 2.4.1 Measurements – East Anglia Network

Methane mixing rations were measured by a network of four sites throughout East Anglia: Tilney-All-Saints Church, Haddenham Church, Weybourne and Tacolneston (Fig. 1). Ellutia GC-FIDs, as described in Sect 2.3.1, were used at Tilney-All-Saints Church, Haddenham Church and Weybourne. Measurement at Haddenham church is described in Sect. 2.3.1, similar systems were arranged at Tilney-All-Saints and Weybourne where inlet were positioned at 25 and 15 m from the

ground, respectively. A Picarro CRDS measured the CH$_4$ mixing ratios in air at Tacolneston at 50 m and 100 m from the ground. Calibration of the Picarro CRDS was done daily for 10 minutes using low (1.93 ppm), target (2.03 ppm) and high (2.74 ppm) CH$_4$ gases calibrated on the World Meteorological Organization (WMO) scale.

### 2.4.2 Model used - InTEM Inversion Modelling

The dispersion model used to represent air flow from potential methane sources to the measurement site is the UK Met

Office's Numerical Atmospheric Modelling Environment (NAME) model (Jones et al., 2007). This is a Lagrangian dispersion model which runs using 3D meteorological fields produced by the UK Met Office's numerical weather prediction model, the Unified Model (UM) (Cullen, 1993). These meteorological fields are available on two resolutions: global (three hourly, 25 km) and UK (hourly, 1.5 km). NAME was run using a combination of both resolutions with the 1.5 km UK fields nested within the global data.



NAME produces a modelled representation of the contributing surface influence (defined as the 100 m above ground level in NAME) to a particular source location over a defined period of time. This is done by releasing chemically-inert particles (10,000 hr⁻¹) from the x, y, z coordinate of a measurement site location. Their movements and geolocation are tracked backwards in time every minute for five days. NAME produces a time-integrated particle density map for each source (units

g s m⁻³), which shows, on a gridded output, what relative contribution each grid square has had over the five day period (Manning et al., 2011). The resolution of this air history map is equal to 1.5 x 1.5 km.

Emissions are inferred in InTEM by using an iterative best fit technique, simulated annealing, which compares the hourly-measured observations with derived modelled observations, based on the NAME InTEM method described in Manning (2003) and Manning et al. (2011). These modelled, or 'pseudo', observations are created by multiplying a simulated

emissions field (g s⁻¹ m⁻³) with a representation of the physical atmospheric processes for each measurement (Eq. 4).

$$emissions\ (g\ s^{-1}\ m^{-2}) \times dilution\ (s\ m^{-1}) = concentration\ (g\ m^{-3}) \qquad (4)$$

The dilution matrix (units s m⁻¹), which links the simulated emission field (g s m⁻³) with the observations (g m⁻²) is produced from the hourly NAME air history maps by dividing by the mass released (g) and then multiplying by a surface area matrix (m²). This dilution matrix is multiplied by the InTEM generated emissions field (both are gridded to the solution grid

resolution).

The two observation time series are quantitatively assessed using a 'least squares' cost function, shown in Eq. 5. For each time step, the difference between the measured ($y_i$) and the pseudo observations (($kx)_i$) is weighted by the total uncertainty (($\sigma_\epsilon^2)_i$), where the uncertainty is defined as the total error estimated in measurement observations, modelling and baselines (Connors et al., in prep). This allows for any potential bias due to highly uncertain observations to be accounted for. InTEM

then iterates for thousands of potential emission fields through the simulated annealing technique to find an optimum result with the lowest cost score (Eq. 5).

$$J(X) = \sum_{i=1}^{m} \frac{(y_i - (kx)_i)^2}{(\sigma_\epsilon)_i^2} \qquad (5)$$

## 3. Model runs

### 3.1 Instantaneous methane emissions – Summer 2015 case study

#### 3.1.1 Near-source - Inverse dispersion modelling

The inversion function of the WindTrax atmospheric dispersion model version 2.0 (Flesch et al., 1995) is used to infer the CH₄ emissions from the Waterbeach landfill using the mixing ratio data collected at Mitchell Hill Farm on the 30th June 2015 and 1st July 2015. Data used as input to WindTrax are: wind speed ($u$, m s⁻¹), wind direction ($WD$, °), temperature ($T$, °C), CH₄ mixing ratio at 4 m ($X$, µg m⁻³), background CH₄ mixing ratio ($X_b$, µg m⁻³), the Monin-Obukhov Length and the surface

roughness. 15-minute-averaged CH₄ mixing ratio data are screened for erroneous values, and data are removed for any periods where wind did not come from the landfill or for high atmospheric stability events, i.e. wind speed, $u < 0.15$ ms⁻¹.



An uncertainty analysis is conducted, where potential variant input values are used in re-run WindTrax scenarios to calculate the resultant change in calculated $CH_4$ emission. These uncertainties are then combined as the square root of the sum of the squares of the individual uncertainties to give an overall uncertainty in emission estimate. The main sources of error are the size of the emission area, as it changed daily, wind speed, the roughness length, and Monin-Obukhov length. The values

used to estimate the uncertainty are from published data.

### 3.1.2 Emissions from middle-distance – Gaussian Plume model

A Gaussian Plume (GP) approach, was used to infer the $CH_4$ emissions from the Waterbeach landfill using the mixing ratio data collected at Haddenham Church on the 30th June 2015 and 1st July 2015. Data used as input to the GP model are: wind speed, wind direction, temperature, $CH_4$ mixing ratio at 4 m, background $CH_4$ mixing ratio and the Pasquill-Gifford

atmospheric stability class. The Pasquill-Gifford stability classes are estimated from calculated values of the Monin-Obukhov length as measured at Mitchell Hill Farm. As with the inverse dispersion modelling approach, 15-minute-averaged data are used and screened for erroneous values, any periods where the prevailing wind did not come from the direction of the landfill or for high atmospheric stability events.

The main uncertainty using the GP approach is in estimating the Pasquill-Gifford atmospheric stability class. The Monin-

Obukhov length is used to assign this value and an uncertainty of ± 7 % was used here because $L$ is calculated using two anemometers each with 5 % uncertainty. Other sources of uncertainty were in the instruments used to measure $CH_4$ mixing ratio and temperature, with uncertainty ranges discussed in Sect. 2. In addition to these sources, a potentially important, yet unquantifiable uncertainty could be off-site sources of emission; unlike the inverse dispersion approach, the GP used in the configuration assumes the landfill is the only point source emitter situated 6 km to the south east of the measurement

location and does not take into account other nearby sources, such as emissions from the on-site generator or other sources upwind. However, any significant difference between the emission estimates calculated using the inversion and the GP approaches may usefully serve to indicate the size of emission from the rest of the Waterbeach Waste Management Park and beyond

### 3.2 Annual and seasonal emission estimates

### 3.2.1 Middle-distance – Gaussian Plume model

The GP approach is described above. Data used as input to GP model are: wind speed, wind direction, temperature, $CH_4$ mixing ratio, background $CH_4$ mixing ratio and the Pasquill atmospheric stability class. Hourly data are used and screened for erroneous values, any periods where wind did not come from the landfill or for high atmospheric stability events.

As with the case study in 3.1.1, the main source of error used as input for the GP approach is the size of the uncertainty in

estimating the Pasquill-Gifford atmospheric stability class. The study also includes the instrument precision and wind speed and temperature uncertainties as derived from the NAME model. Also, we assume the landfill is the only point source



emitter 6 km to the south east and does not take into account other nearby sources, such as emissions from the on-site generator and further upwind.

### 3.2.2 Landscape - InTEM Inversion Model

InTEM was run using data from all four measurement sites (Fig. 1) between $1^{st}$ June 2013 and $31^{st}$ May 2014. Repeating the
inversion method gives slightly different cost scores and emission totals due to the stochastic nature of the changes made during the simulated annealing process (Manning et al., 2011). For this study, InTEM was repeated 25 times as this resulting in consistent methane emission estimates, standard deviations and cost score.

Methane emissions are produced on a solution grid of varying spatial resolution. This resolution is determined using the NAME air history maps and the National Atmospheric Emissions Inventory (NAEI) for methane. Surface regions which
have a larger influence on the observation sites and have a large emission in the NAEI produce boxes at a higher spatial resolution. The smallest resolution allowed for the emission grid is set equal to the NAME grid resolution (1.5 x 1.5 km). The box which contains the Cottenham landfill site is roughly 9 x 4.5 km.

An estimated methane baseline mixing ratio is calculated to represent the methane mixing ratio that would have been measured at a given site in the absence of emissions from within the dispersion domain. A statistical filtering technique
separated methane mixing ratios at each site into eight-time series using the NAME air history maps by wind direction. A rolling $18^{th}$ percentile spanning one week is then passed through each time series. Sensitivity analysis shows this baseline produces emission results with consistently stable emissions with the lowest cost score of all baselines tested.

The uncertainty estimates used within InTEM reflect the variability of the resulting emission estimates. Uncertainty is defined as the total of the calibration gas uncertainty range, the GC instrument precision and the standard deviation within
the hourly observation, plus a default mixing ratio of 5 ppb to represent uncertainty with the baseline and dispersion modelling. For a more detailed description of the measurement sites and the InTEM setup please refer to Connors et al. (in prep).

### 4. Results

#### 4.1 Isotopic methane measurements

Several large $CH_4$ plumes were measured by the GC-FID in Haddenham Church on the $11^{th}$ February 2015 (Fig. 2) during a wind event from the south east ranging from background, $c.$ 1900 ppb, to a maximum mixing ratio of 2460 ppb. Air samples collected in Tedlar bags at the same time at the same location and analysed later for $CH_4$ mixing ratio using a Picarro CRDS at RHUL show good agreement in measurement between the GC-FID and Picarro CRDS.

The $\delta^{13}C$ isotopic signature of the source contributing to excess methane over background can be calculated using the
Keeling plot approach (e.g. Zazzeri et al., 2015). This is a plot of $1/CH_4$ (ppm) vs measured isotopic signature for each sample. The intercept of the correlation line fit where $1/CH_4 = 0$ closely approximates the source signature. The Keeling plot




of the air samples taken at Haddenham Church between 0600 and 1400 hours on 11$^{th}$ February 2015 estimates the $\delta^{13}$C isotopic signature at -58.3 ‰ (Fig. 3). The typical $\delta^{13}$C isotopic signature value for a landfill in the south east of the UK has been estimated to be -58 ± 3 ‰ (Zazzeri et al., 2015), which is very different from other possible local source signatures such as fossil fuels or combustion. This strongly suggests that the air measured at the church has come from a landfill. Air

samples were taken closer to the landfill, 10 m from the active site.

**4.2 Estimating methane emissions – Case study June 2015**

The average $CH_4$ emission for the Waterbeach landfill in July based on near source $CH_4$ measurements used in WindTrax is estimated at 565 µg m$^{-2}$ s$^{-1}$ (453 kg hr$^{-1}$). In general, emissions on the 30$^{th}$ June (average = 256 µg m$^{-2}$ s$^{-1}$) are ten times lower than those on the 1$^{st}$ July (average = 2840 µg m$^{-2}$ s$^{-1}$), corresponding to less stable conditions and lower atmospheric pressure

on the 1$^{st}$ (Fig. 4). The maximum emission is estimated at 18700 µg m$^{-2}$ s$^{-1}$ at 1215 UTC on the 1$^{st}$ July.
A range of scenarios were run in WindTrax to investigate the uncertainty in $CH_4$ emissions caused by the $CH_4$ measurement, the wind speed measurement, estimating the roughness length and estimating the Monin-Obukhov length. Realistic uncertainty in the Monin-Obukhov length and instrument uncertainty for the $CH_4$ measurement have little effect on the emission estimate. Uncertainty in estimating the emission area and roughness length have a noticeable effect on $CH_4$

emission, resulting in an uncertainty of ± 3 % and ± 4 % on modelled $CH_4$ emissions, respectively. WindTrax has the greatest response to the uncertainty in estimating wind speed, resulting in an emission uncertainty of ± 19 %. The overall uncertainty in $CH_4$ emission, calculated as the root of the sum of each component squared, is estimated at ± 20 % (Table 2).
The methane emissions calculated using the WindTrax model can be compared with those calculated by a Gaussian plume model using the same measurements. As with WindTrax, the emissions on the 30$^{th}$ June (average = 408 µg m$^{-2}$ s$^{-1}$) are lower

than those on the 1$^{st}$ July (average = 1270 µg m$^{-2}$ s$^{-1}$). However, the difference in emissions is not as large (Fig. 5). The maximum emission is estimated at 2590 µg m$^{-2}$ s$^{-1}$ at 1215 UTC on the 1$^{st}$ July, which suggests that the Gaussian plume approach measures a more mixed emission than the inversion dispersion model.
A range of scenarios were also configured using the Gaussian plume approach to reflect uncertainty in $CH_4$ measurement, wind speed measurement, temperature measurement and the Monin-Obukhov length (Table 3). Changing the Monin-

Obukhov length had no detectable effect on the emission estimate because the change in $L$ is not enough to vary the assigned Pasquill-Gifford Stability class use in the emission calculation. Varying the temperature and wind speed had little effect on $CH_4$ emission and resulted in an uncertainty of ± 1 % and ± 5 % on modelled $CH_4$ emissions, respectively. The uncertainty in estimating $CH_4$ emissions caused by the instrument precision is the greatest source of uncertainty and results in an uncertainty of the emission estimate of ± 22 %. The overall uncertainty in $CH_4$ emission, calculated as the root of sum of

each component squared, is estimated to be ± 23 %.





### 4.3 Annual and seasonal emission estimates

Methane emissions from the landfill at the Waterbeach Waste Management Park were calculated using 1171 hourly averaged $CH_4$ mixing ratio data measured at Haddenham Church between July 2012 and June 2015. The GP model can only be used to calculate the emission when the wind is blowing from the SE (i.e. from the landfill). For this particular time series, the

wind was only from the SE for 1171 hours. Meteorological data from the Unified Model analyses are used to calculate the Pasquill-Gifford stability class. When applied in the Gaussian Plume model, the monthly average $CH_4$ emission for July is estimated at 616 kg hr$^{-1}$, in reasonable agreement with the estimates of 453 and 641 kg hr$^{-1}$ of the WindTrax inverse dispersion and Gaussian Plume models using measured meteorological data. Emissions for all months are shown in Table 4.

In general, $CH_4$ emission rates are higher during the winter months and lower during the summer months (Fig. 6). During

the winter months (December, January, February) $CH_4$ emission from the landfill is estimated at 1860 kg hr$^{-1}$ (441 µg m$^{-2}$ s$^{-1}$), whereas in the summer months (June, July, August) the $CH_4$ emission drops to more than half to 930 kg hr$^{-1}$ (207 µg m$^{-2}$ s$^{-1}$). Variability in emissions is also larger in winter than in summer. The mean annual emission, calculated as the sum of the monthly mean emissions, is estimated at 11.6 Gg yr$^{-1}$.

As before scenarios were ran using the GP approach to reflect variability in instrument precision, wind speed, temperature

and the Pasquill-Gifford stability class (Table 5). Changing the temperature had no effect on the emission estimate, and instrument precision was a larger source of uncertainty, ± 9 %. However, the effect of instrument precision was smaller than the uncertainty in the case study possibly because the measured mixing ratios are at their lowest during the summer. The calculation of the PGSC and the uncertainty in wind speed were the highest source of uncertainty resulting in variability in $CH_4$ emission of ± 24 % and ± 20 %, respectively. The overall uncertainty in $CH_4$ emission is estimated to be ± 32 %.

### 4.3.2 InTEM Inversion Model Methane Emission Estimates

The average annual $CH_4$ emission from the landfill calculated using ~24,000 hourly averaged $CH_4$ mixing ratio data measured by the East Anglia network (Fig. 1) and NAME modelled met data in the InTEM model is estimated at 13.7 Gg yr$^{-1}$ (Table 4). The emission estimate was calculated from the average $CH_4$ emission 19.9 µg m$^{-2}$ s$^{-1}$ calculated for an area of $2.17 \times 10^7$ m$^2$. The standard deviation of the $CH_4$ emission for 25 repeat runs of the InTEM model is $1.8 \times 10^{-5}$ g s$^{-1}$ m$^{-2}$ (91

25   %).

### 5. Discussion and Conclusions

The data presented in this paper gives the first comparison of methane emissions from a working landfill calculated using three models at different scales: (a) near-source < 1 km (WindTrax); mid-distance 1-7 km (Gaussian Plume); and far field 7 – 70 km (InTEM). Near-source measurements were taken 300 m to the north west of the Waterbeach Waste Management

Park, Cambridgeshire on the 30[th] June and 1[st] of July 2015. Mid-distance measurements were taken from Haddenham





Church, 7 km north west of the landfill, between July 2012 and July 2015. Far-field measurements were taken throughout East Anglia, ranging from 7 km to 100 km from the landfill, between July 2012 and July 2015.

After using $^{13}CH_4$ signatures to confirm that the source of the large $CH_4$ mixing ratios is a nearby landfill, $CH_4$ emissions estimated using near-source measurements are 453 kg hr$^{-1}$ in June/July 2015 and agree within associated uncertainties when

compared to the mid-distance emission estimates of 641 kg hr$^{-1}$. From the limited observation period, we also observe greater variability in emissions using the near-source method, in accord with the finding of Riddick et al (2016) that suggest that near-source estimates can be affected by the heterogeneous nature of the landfill. Using mid-distance measurement throughout the year we estimate the annual $CH_4$ emissions from the site to be 11.6 Gg yr$^{-1}$ which is comparable to the $CH_4$ emission estimate as calculated using the InTEM inversion method of 13.7 Gg yr$^{-1}$.

The $CH_4$ emissions from this landfill site are seasonal with the largest emissions during the winter months (February 2160 kg hr$^{-1}$) and the lowest emissions during the summer months (616 kg hr$^{-1}$). This may be linked to the seasonal cycle in environmental conditions as there is an inverse relationship between $CH_4$ emission and temperature. The temperature relationship may be explained by the increased activity of methanotrophic bacteria in the top layers of landfill as the temperature increases.

The $CH_4$ emissions from this landfill site are seasonal with the largest emissions during the winter, colder months (February: 2160 kg hr$^{-1}$) and the lowest emissions during the summer, warmer months (616 kg hr$^{-1}$). This is explained by the following mechanism (Börjesson and Svensson, 1997). The temperature within the landfill is relatively stable so that the sub-surface production of $CH_4$ is roughly constant. In summer when the surface temperature is higher, the activity of methanotrophic bacteria in the top layers of landfill is enhanced, so that the net emission into the atmosphere is reduced. Our measurements

are the first off-site measurements to demonstrate this and so are not susceptible to the sampling uncertainties associated with chamber techniques.

The $CH_4$ emission estimate made by this study of 13.7 Gg yr$^{-1}$ from this site is an important contribution to the waste component (714 Gg yr$^{-1}$) of the 2014 total UK $CH_4$ emission inventory (2,157 Gg yr$^{-1}$; NAEI, 2016). We estimate the 13.7 Gg yr$^{-1}$ emitted is produced from the 400 Gg of total waste processed each year at the site (AMEY, 2016). The inferred $CH_4$

emission to waste ratio at this site is lower (0.034) than the current UK ratio (0.045), where 1.0 Tg $CH_4$ yr$^{-1}$ (EC-JRC/PBL, 2010) is emitted from 22 Tg of solid waste disposed in landfill (UK Gov, 2016). This may be the result of differing environmental and management factors, such as differing mass fractions for each decomposing waste category (Jung et al., 2010), movement of landfill leachate (Attenborough et al., 2002) and site specific weather conditions (Maurice & Lagerkvist, 2004; Scheutz et al., 2004; Xu et al., 2014). Alternatively, $CH_4$ emissions from new landfills which include a

high component of recycling are currently overestimated.

The agreement between the mid-distance estimates and the NAME InTEM inversion model shows that reasonably dense measurement networks that provide data for regional inversion models can be used to identify emission hotspots within the network and even to quantify their emissions hotspots. Once potential hotspot emission sources have been identified, year-round measurements coupled to a relatively simple Gaussian plume model could be used to estimate the annual average and




any seasonality in the $CH_4$ emissions. As lower cost sensors become available, a cost-effective system to monitor point source emissions should become available.

**Acknowledgements**

This project was supported by the UK Natural Environment Research Council (NERC) through the Greenhouse gAs Uk and

Global Emissions (GAUGE) project on grant number NE/K002570/1. We also thank the Department of Environment, Farming and Rural Affairs and the Royal Society for seed funding and NERC for additional support through grants NE/G014655/1, NE/J006246/1 and a PhD studentship for Sarah Connors.  Special thanks to the owners of Mitchell Hill Farm, Cottenham and to Holy Trinity church, Haddenham for allowing us to site our instruments on their land.

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




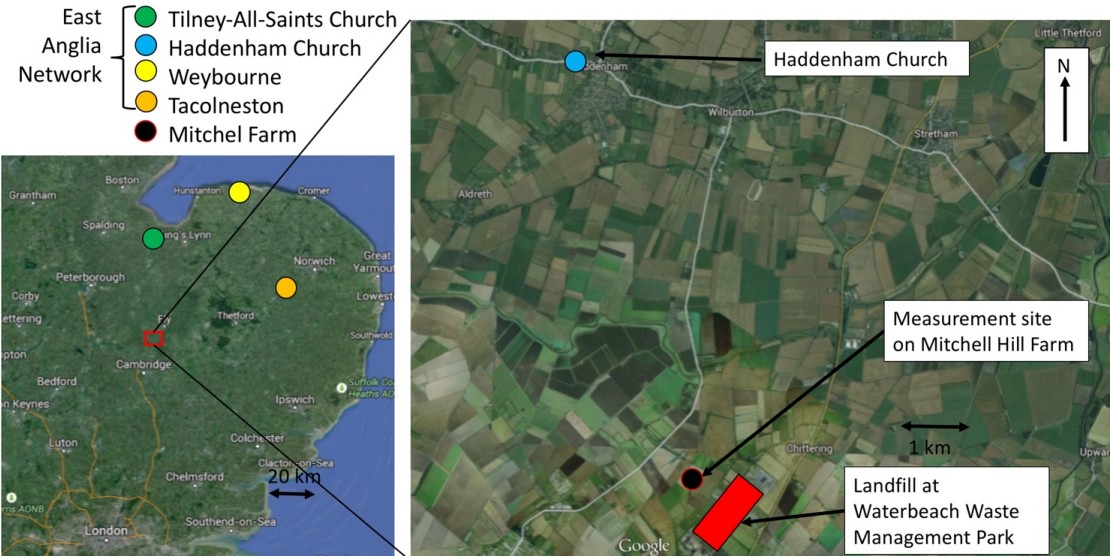

**Figure 1. Location of the East Anglia Measurement Network (Tilney-All-Saints Church, Haddenham Church, Weybourne & Tacolneston), landfill at the Waterbeach Waste Management Park and the measurement site at Mitchells Hill farm, Cambridgeshire. The map was taken on 23$^{rd}$ July 2015 (Google Earth, 2015).**



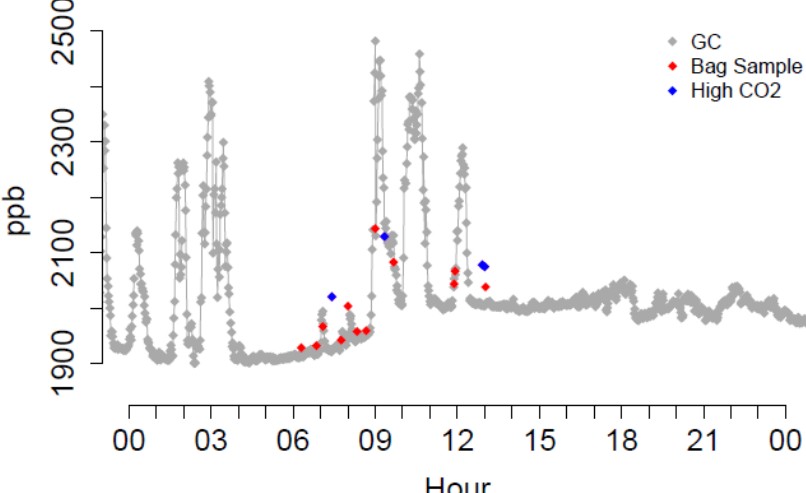

**Figure 2 Methane mixing ratios measured by the GC-FID in Haddenham Church on 11th February 2015 are presented in grey. Matching methane mixing ratios collected in Tedlar bags on the 11th February 21015 and analysed on the 20th February 2015**
5 **using a Picarro CRDS at Royal Holloway University of London are presented as red points.**





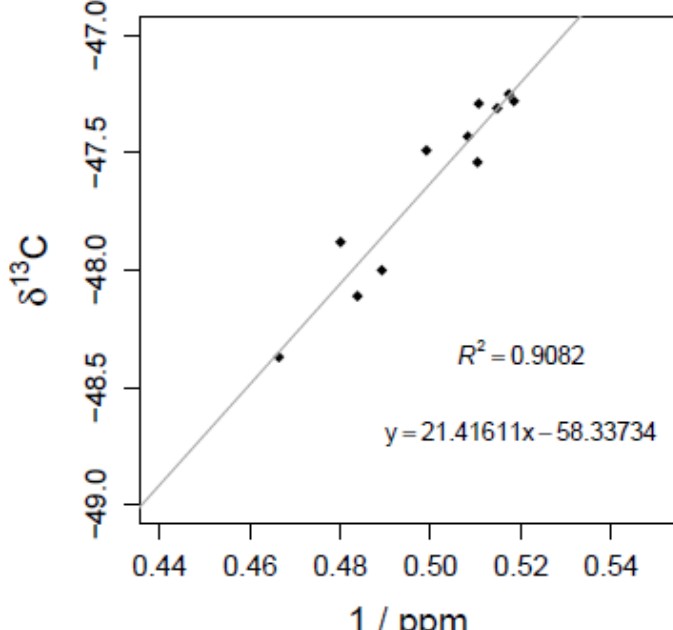

**Figure 3 Keeling plot of the air samples taken at Haddenham Church between 0600 and 1400 hours on 11th February 2015.**




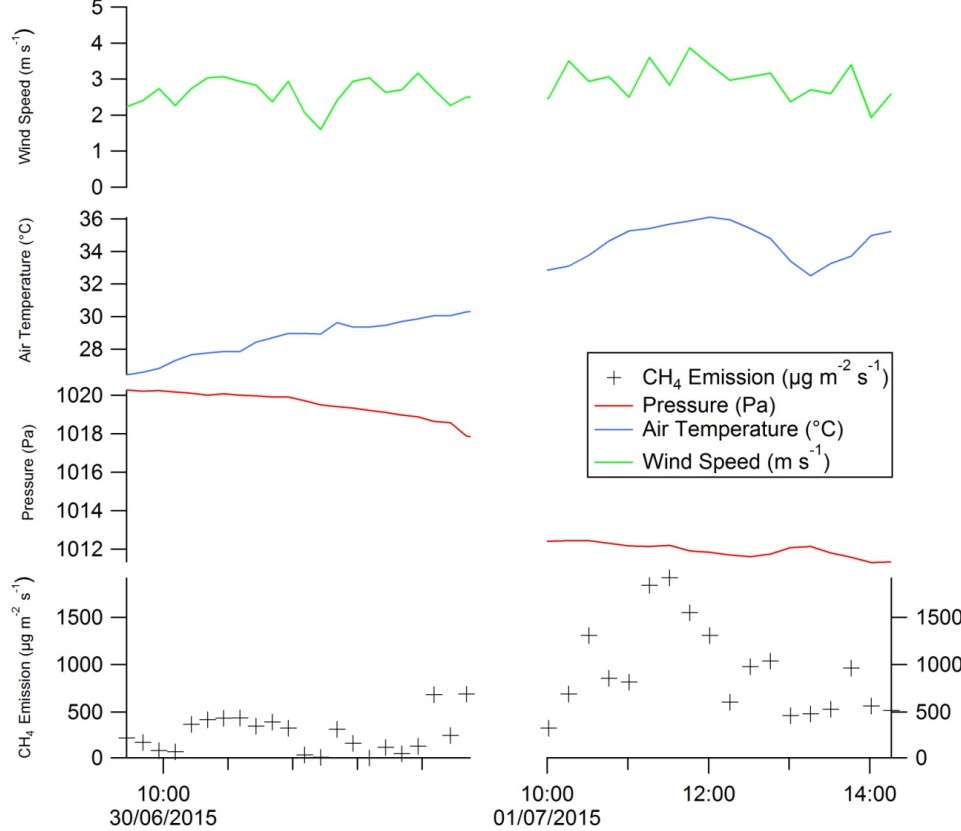

**Figure 4 Measured wind Speed (top), measured temperature (middle-top), measured pressure (middle-bottom) and methane emission rate as calculated by the WindTrax atmospheric dispersion model (bottom) from data collected at Mitchell Hill Farm, Cottenham from the landfill at the Waterbeach Waste Management Park on the 30th June and 1st July 2015.**





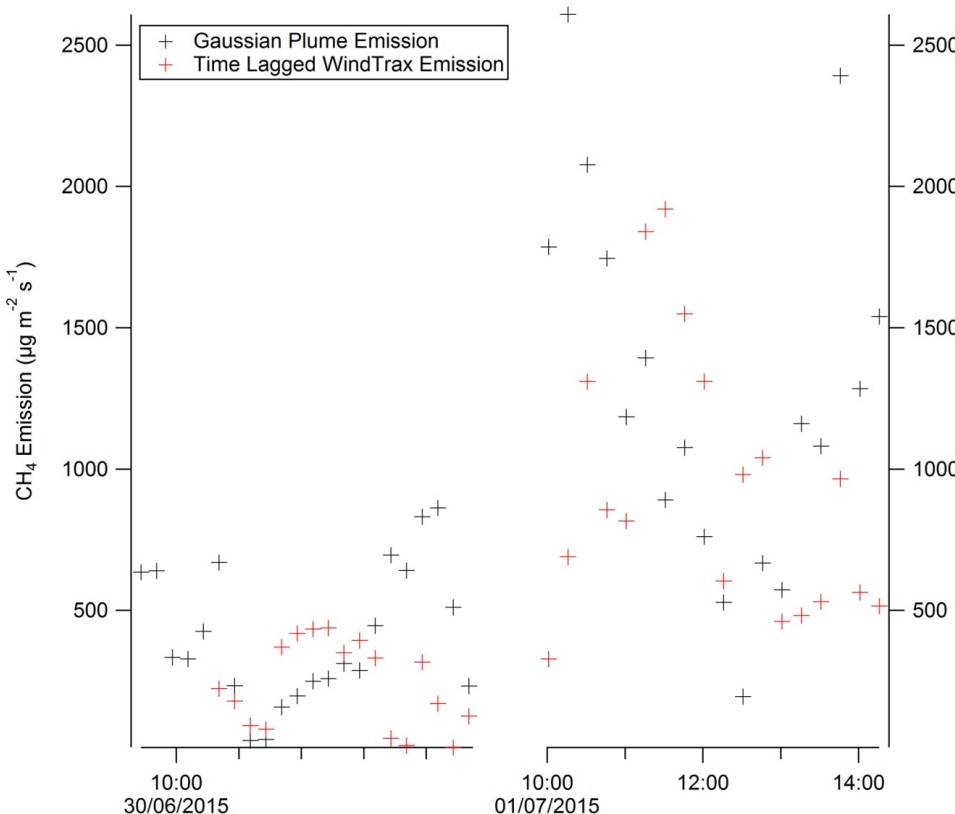

**Figure 5 Methane emission rate as calculated by the Gaussian Plume modelling approach (black crosses) and the WindTrax atmospheric dispersion model (red crosses) from data collected at Mitchell Hill Farm, Cottenham from the landfill at the Waterbeach Waste Management Park on the 30th June and 1st July 2015.**





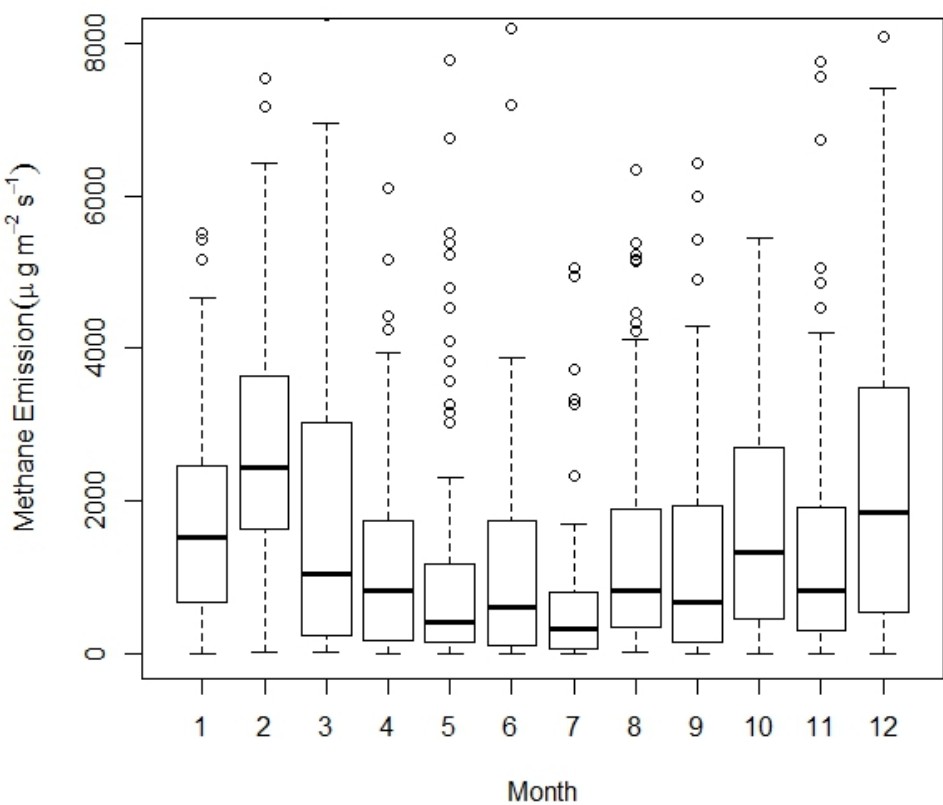

**Figure 6 Box plot of hourly emissions calculated using the Gaussian Plume modelling approach showing the monthly variability in methane emissions using data from 2012 to 2014.**



**Table 1 Summary of methods used to calculate methane emission estimates from a landfill at different scales: near-source, middle-distance and landscape.**

| Scale | Measurement location (Fig. 1) | Measurement method | Meteorological data | Model used to calculate emission |
|---|---|---|---|---|
| Near-source | Mitchel's Farm, Cottenham, Cambridgeshire | Los Gatos UGGA (Sect. 2.2.1) | In-situ at Mitchells' Farm (Sect. 2.2.2) | WindTrax Inverse Dispersion Model (Sect. 2.2.3) |
| Middle-distance | Haddenham Church, Cambridgeshire | Ellutia 200 Series GC-FID (Sect. 2.3.1) | NAME Model Runs (Sect. 2.4.2) | Gaussian Plume (Sect. 2.3.3) |
| Landscape | East Anglia measurement Network | Ellutia 200 Series GC-FID (Sect. 2.3.1) Picarros CRDS (Sect. 2.4.1) | NAME Model Runs (Sect. 2.4.2) | InTEM Model (Sect. 2.4.2) |



**Table 2 Uncertainty analysis conducted on the case study (June 2015) methane emission from the landfill at the Waterbeach Waste Management Park as calculated within the WindTrax atmospheric dispersion model.**

| Variable | Value used | Average Emission ($\mu g\ m^{-2}\ s^{-1}$) | Uncertainty (%) |
|---|---|---|---|
| Baseline | | 565 | |
| Monin-Obukhov Length | ± 7% | 563 | ± 0.3 |
| Precision Roughness Length | ± 7% | 588 | ± 4 |
| $CH_4$ Instrument | ± 0.01% | 567 | ± 0.3 |
| Wind Speed Measurement | ± 5% | 671 | ± 19 |
| Emission area | ± 20% | 547 | ± 3 |
| | | Total | ± 20 |



**Table 3 Uncertainty analysis conducted on the case study (June 2015) methane emission from the landfill at the Waterbeach Waste Management Park as calculated within the Gaussian Plume modelling approach.**

| Variable | Value used | Average Emission ($\mu g\ m^{-2}\ s^{-1}$) | Average Emission ($kg\ hr^{-1}$) | Uncertainty (%) |
|---|---|---|---|---|
| Baseline | | 800 | 641 | |
| $CH_4$ Instrument Precision | ± 0.5% | 973 | 781 | 22 |
| Wind Speed Measurement | ± 5% | 840 | 674 | 5 |
| Monin-Obukhov Length | ± 7% | 800 | 641 | 0 |
| Temperature Measurement | ± 5% | 795 | 638 | 0.4 |
| | | | Total | 23 |





**Table 4 Methane emission estimates from the landfill at the Waterbeach Waste Management Park as calculated by the WindTrax and Gaussian Plume approaches for the case study (June 2015) and the annual estimates for the Gaussian Plume and InTEM inversion modelling approach for 2012 – 2104.**

| Month | Case Study | | Annual Estimate | |
|---|---|---|---|---|
| | Inverse dispersion | Gaussian Plume | Gaussian Plume | InTEM |
| | $(kg\ hr^{-1})$ | $(kg\ hr^{-1})$ | $(kg\ hr^{-1})$ | |
| January | | | 1370 | |
| February | | | 2160 | |
| March | | | 1580 | |
| April | | | 111 | |
| May | | | 830 | |
| June | | | 1070 | |
| July | $453 \pm 20\%$ | $641 \pm 23\%$ | 616 | |
| August | | | 1100 | |
| September | | | 1480 | |
| October | | | 1350 | |
| November | | | 1210 | |
| December | | | 2040 | |
| | | Total Emission $(Gg\ yr^{-1})$ | $11.6 \pm 32\%$ | $13.7 \pm 91\%$ |





**Table 5 Uncertainty analysis conducted on the annual methane emission from the landfill at the Waterbeach Waste Management Park as calculated within the Gaussian Plume modelling approach**

| Variable | Value used | Average Emission ($\mu g\ m^{-2}\ s^{-1}$) | Average Emission ($kg\ hr^{-1}$) | Uncertainty (%) |
|---|---|---|---|---|
| Baseline | | 1650 | 1320 | |
| $CH_4$ Instrument Precision | ± 0.5% | 1790 | 1440 | 9 |
| Wind Speed Measurement | ± 20% | 1980 | 1590 | 20 |
| PGSC | ± 1 SC | 1490 | 1200 | 24 |
| Temperature Measurement | ± 20% | 1640 | 1320 | 1 |
| | | | Total | 32 |