# Peer review of "Estimating the size of a methane emission point-source at different scales: from local to landscape"

_Atmospheric Chemistry and Physics, 2016_

## Referee Comment (RC1) · Anonymous Referee #1 · 31 Jan 2017

The comment was uploaded in the form of a supplement:
http://www.atmos-chem-phys-discuss.net/acp-2016-963/acp-2016-963-RC1-supplement.pdf

---

## Referee Comment (RC2) · Anonymous Referee #2 · 7 Feb 2017

**General comments**

This is an interesting paper discussing observations of atmospheric methane at different scales (near-source, middle-distance, and landscape), and their value for quantifying a methane emission point-source (using specialized estimation methods). My main problem in this paper is the conclusion that the landscape inverse modeling approach can be used to identify point sources. The inversion method lacks details and the discussion is somewhat superficial. I think OSSEs would be required to determine the ability of observations at the landscape scale to constrain emission hotspots.

**Specific comments**

P6, L9-10: "The standard deviation of the lateral ( $\sigma$ y, m) and vertical ( $\sigma$ z, m) mixing ratio distribution are calculated from the stability class of the air (Pasquill, 1974)." So

what are the values for the standard deviation used in this paper?

P7, L19: "This allows for any potential bias due to highly uncertain observations to be accounted for." I don't see how the bias would be accounted for.

P9, L14-15: "A statistical filtering technique separated methane mixing ratios at each site into..." What is this statistical filtering?

P9, L16: Why "18th percentile"? Why not 10th or 25th?

P9, L21: "For a more detailed description of the measurement sites and the InTEM setup please refer to Connors et al. (in prep)." I think more details about the InTEM setup should be given. For example, what prior constraints or regularization do you use? This is crucial for an inversion.

P12, L4: "...using near-source measurements are 453 kg hr-1 in June/July 2015..." I thought the near-source measurements cover only two days? This looks like two-month data.

P12, L15-20: Table 4 shows the lowest emissions month is in April (111 kg/hr). I am not very convinced that seasonality is due to temperature. Does stability class in the Gaussian plume approach play a role?

P12, L33-34: I am not convinced by this conclusion. See my general comments.

**ACPD**

---

## Author Comment (AC1) · 23 Mar 2017

Please see attached pdf (acp-2016-963_Reviewer1) for our detailed response.

Please also note the supplement to this comment:
http://www.atmos-chem-phys-discuss.net/acp-2016-963/acp-2016-963-AC1-supplement.pdf
* * *
[Figure]
* * *
Ms. Ref. No.: acp-2016-963

Title: Estimating the size of a methane emission point-source at different scales: from local to landscape

Department of Civil and Environmental Engineering
Princeton University
E320 Engineering Quad
Princeton
NJ

Email: sriddick@princeton.edu

23rd March 2017

Dear Editor,

We would like to thank the referee #1 for their comments. As suggested, we have ammneded the claims made by the paper regarding the power of the inversion model to calculate point sources and have provided details of a new publication that provides more detail on the InTEM modeling.

Please find our detailed responses below.

Yours sincerely,

Stuart Riddick (corresponding author)

and co-authors: Sarah Connors, Andrew Robinson, Alistair Manning, Pippa Jones, David Lowry, Euan Nisbet, Robert Skelton, Grant Allen, Joseph Pitt and Neil Harris

**Fig. 1.** Response
Supplementary Material Section 1

The standard deviation of the lateral ($\sigma_y$, m) and vertical ($\sigma_z$, m) mixing ratio distribution calculated from the stability class of the air (Pasquill, 1974).

| Stability Class | Day | | | Night | |
|---|---|---|---|---|---|
| Wind Speed (m s$^{-1}$) | Strong | Mod | Light | Overcast | Clear |
| 2 | a | a | b | | |
| 3 | b | b | c | e | f |
| 4 | b | c | c | d | e |
| 5 | c | c | d | d | d |
| 6 | c | d | d | d | d |

| | sigz | |
|---|---|---|
| Stability Class | a | b |
| A | 0.0002539 | 2.089 |
| B | 0.04936 | 1.114 |
| C | 0.1154 | 0.9109 |
| DD | 0.7368 | 0.5642 |
| DN | 1.297 | 0.4421 |
| E | 0.9204 | 0.4805 |
| F | 1.505 | 0.3662 |

| | sigy | |
|---|---|---|
| Stability Class | c | d |
| A | 0.495 | 0.873 |
| B | 0.31 | 0.897 |
| C | 0.197 | 0.908 |
| DD | 0.122 | 0.916 |
| DN | 0.122 | 0.916 |
| E | 0.0934 | 0.912 |
| F | 0.0625 | 0.911 |

**Fig. 2.** Supplementary Material Section 1

Supplementary Material Section 2

[Figure]

Scatterplot of posterior enhancements vs. observed enhancements.

**Fig. 3.** Supplementary Material Section 2

---

## Editor Comment (EC1) · E. Harris (Editor) · 12 May 2017

Please find below an additional referee comment:

A plot included in the response to reviewer #2 appears problematic. This is the plot under "Supplementary Material Section 2: Scatterplot of posterior enhancements vs. observed enhancements".

There are three problems with this plot. The first two are matters of presentation, and the third is scientifically substantive.

First: these are not enhancements, they are concentrations. In the inverse modeling literature an enhancement is the concentration of the constituent at the observation site minus the background. A plot of modeled vs observed enhancements allows one

to separate the variability in the background from the variability in the influence of local emissions. When one plots modeled vs. observed concentrations, a well modeled background will hide problems with the model of the emissions.

Second: the points in this plot are so dense that they are indistinguishable. There are plotting strategies that can ameliorate this, like coloring by point density or contouring.

Third, and most importantly, I think this plot indicates a problem in the model. There appears to be heteroscedasticity in the model residuals. This is likely due to the fact that the background was not subtracted. The variation in the model and observed background is likely falling along the 1-1 line, and the enhancements are likely falling off of it. There appears to be a number of highly influential outliers.

A plot of modeled vs observed enhancements is generally considered among the most basic diagnostic plots for an inverse analysis. It would be good to see this type of plot for every site and every model. It isn't clear for which model this plot is showing.

---

## Author Comment (AC2)

Ms. Ref. No.: acp-2016-963

Title: Estimating the size of a methane emission point-source at different scales: from local to landscape

Department of Civil and Environmental Engineering
Princeton University
E320 Engineering Quad
Princeton
NJ

Email: sriddick@princeton.edu

23rd March 2017

Dear Editor,

We would like to thank the referee #2 for their comments. Please find our detailed responses below.

Yours sincerely,

Stuart Riddick (corresponding author)

and co-authors: Sarah Connors, Andrew Robinson, Alistair Manning, Pippa Jones, David Lowry, Euan Nisbet, Robert Skelton, Grant Allen, Joseph Pitt and Neil Harris

Reviewer #2

| My main problem in this paper is the conclusion that the landscape inverse modeling approach can be used to identify point sources. The inversion method lacks details and the discussion is somewhat superficial. I think OSSEs would be required to determine the ability of observations at the landscape scale to constrain emission hotspots. | We have refocused the paper and made the point that distinct emission sources can be observed within an emission landscape. We suggest that landscape inversion models can be used to identify emission hot-spots within an emission landscape. | Page 1 L25 the following has been removed: "is in good agreement with more labour-intensive near-source approaches and"

Page 1 L26 the following has been removed: "to provide high-quality emission estimates"

Page 12 L31 the following was removed: "agreement between the mid-distance estimates and the" and "that provide data for regional inversion models"

Page 13 L19 the following was added: "output from"

Page 12 L32 the following was removed: "the network and even to quantify their emissions hotspots" |
| --- | --- | --- |

| | | Page 13 L20 the following was added "an emission landscape" |
|---|---|---|
| P6, L9-10: "The standard deviation of the lateral ($\sigma y$, m) and vertical ($\sigma z$, m) mixing ratio distribution are calculated from the stability class of the air (Pasquill, 1974)." So what are the values for the standard deviation used in this paper? | The values used can be found in the Supplementary Material Section 1. | |
| P7, L19: "This allows for any potential bias due to highly uncertain observations to be accounted for." I don't see how the bias would be accounted for. | High methane concentration values seen at Haddenham are usually short lived and appear as peaks lasting only a few hours (max). They usually occur at nighttime and, as the isotopic analysis shows, probably come from a landfill, which is an intermittent of methane. These are therefore more uncertain. The values would have a relatively high cost score at these times. So, by including the hourly SD into the uncertainty calculation this helps to de-weight the | |

| | | |
|---|---|---|
| | large concentrations, which have higher uncertainty, from increasing the overall cost score. | |
| P9, L14-15: "A statistical filtering technique separated methane mixing ratios at each site into. . ." What is this statistical filtering? | See comment above. | |
| P9, L16: Why "18th percentile"? Why not 10th or 25th? | This percentile is used as a result of sensitivity analysis showing that the resulting InTEM inversion results produced the lowest cost scores and therefore means the emissions produced are closer to the measured observations than any other percentiles tested. I tested from the $5^{th}$ to the $45^{th}$. Sensitivity analysis shows this baseline produces emission results with consistently stable emissions with the lowest cost score of all baselines tested. | |
| P9, L21: "For a more detailed description of the | A new paragraph is included to make the link to the | Text added at P9 L12: |

| | | |
|---|---|---|
| measurement sites and the InTEM setup please refer to Connors et al. (in prep).” I think more details about the InTEM setup should be given. For example, what prior constraints or regularization do you use? This is crucial for an inversion. | InTEM setup described in Connors et al. (in prep.), Sarah Connors' thesis and the new information in the supplementary material clearer.

This inversion does not use a prior, like the other studies referenced here. Priors are not essential but they are more commonly used than not. It uses a cost function similar to a least-squares approach. Bayesian cost-functions use priors and the analysis could (and probably should) be repeated with a Bayesian CF to comparison and a better assessment of errors. | “The results presented here are taken from a study developing a method to estimate regional CH4 emissions in East Anglia (Connors et al., in prep.). More details on the measurements sites, the inversion set-up used for InTEM, the diagnostics used and the emission uncertainties can be found there and in Connors (2015). The main points for the purposes of this paper are summarised below and in the Supplementary Material.” |
| P12, L4: “. . .using near-source measurements are 453 kg hr-1 in June/July 2015. . .” I thought the near-source measurements cover only two days? This looks like two-month data. | Corrected as suggested | Added at p12 L16:
“30th June and 1st July 2015” |
| P12, L15-20: Table 4 shows the lowest emissions month is | This was typo and should be 1110 kg/hr and has been | |

| | | |
|---|---|---|
| in April (111 kg/hr). I am not very convinced that seasonality is due to temperature. Does stability class in the Gaussian plume approach play a role? | corrected. The response of $CH_4$ emission from landfill to temperature is well documented and a result of methanotrophic bacteria becoming more active during the summer months. | |
| P12, L33-34: I am not convinced by this conclusion. See my general comments. | | Added at P12 L19:
"We suggest that the agreement in emissions estimates between the near-source and middle-distance methods indicate that a Gaussian plume approach can be used to estimate emissions up to 7 km from a relatively large source. However, this may be an upper estimate of the distance that this approach is effective as the fetch between the source and detector was relatively flat and a more aerodynamically complex landscape may reduce the model's efficacy."

At P12 L25:
"Our results suggest that larger emission hot-spots can be detected within the |

| | | emission landscape generated by an inversion model. However, we would suggest that future sensitivity studies should be conducted to estimate the size of emission hot-spots within a landscape where the source is farther from a measurement site used as input to the inversion model." |
|---|---|---|
| | | |

---

## Author Comment (AC3)

[Figure]

**Figure SM2.1 An example of the time-series of observations, background values and posterior enhancement as calculated by the InTEM model from July 2013 to August 2013 for the measurement site in Haddenham, Cambridgeshire.**

[Figure]

**Figure SM2.2 Scatterplot of posterior enhancements vs. observed enhancements as calculated by the InTEM model for the Haddenham, Cambridgeshire.**

[Figure]

**Figure SM2.3 Distribution of hourly methane emissions from the landfill as calculated using the Gaussian Plume model using methane concentration data collected at Haddenham church. The data includes emission estimates from July 2012 to July 2014. The red line represents the emission from the landfill as calculated by the InTEM model.**

---

## Author Response (AR1)

Ms. Ref. No.:  acp-2016-963

Title: Estimating the size of a methane emission point-source at different scales: from local to landscape

Department of Civil and Environmental Engineering
Princeton University
E320 Engineering Quad
Princeton
NJ

Email: sriddick@princeton.edu

23$^{rd}$ March 2017

Dear Editor,

We would like to thank the referee #1 for their comments. As suggested, we have ammneded the claims made by the paper regarding the power of the inversion model to calculate point sources and have provided details of a new publication that provides more detail on the InTEM modeling.

Please find our detailed responses below.

Yours sincerely,

Stuart Riddick (corresponding author)

and co-authors: Sarah Connors, Andrew Robinson, Alistair Manning, Pippa Jones, David Lowry, Euan Nisbet, Robert Skelton, Grant Allen, Joseph Pitt and Neil Harris

Reviewer #1

| I would suggest that the authors are left with two avenues of recourse. (1) They could repeal the claim on the power of the landscape inversion. This would leave a very nice paper focused on the emissions from the Waterbeach waste management park, and I would happily support its publication (with the other, smaller concerns in this review addressed). (2) They could conduct a well-designed OSSE, testing their claim vigorously and assuaging my concerns directly. | In light of the reviewers comments we would like to pursue avenue (1), where we have refocused the paper and made the point that distinct emission sources can be observed within an emission landscape. We suggest that landscape inversion models can be used to identify emission hot-spots within an emission landscape. | Page 1 L25 the following has been removed: "is in good agreement with more labour-intensive near-source approaches and"

 Page 1 L26 the following has been removed: "to provide high-quality emission estimates"

 Page 12 L31 the following was removed: "agreement between the mid-distance estimates and the" and "that provide data for regional inversion models"

 Page 13 L19 the following was added: "output from"

 Page 12 L32 the following was removed: "the network and even to quantify their emissions hotspots" |
| --- | --- | --- |

| | | Page 13 L20 the following was added "an emission landscape" |
|---|---|---|
| The introduction gives a very good discussion of methane emissions from landfills, but not an adequate discussion of the issues surrounding the degree to which point sources can be resolved by atmospheric measurements at different scales (i.e., near-source, middle-distance, and landscape). | Changed as suggested, but we are not aware of many papers. | The following was added at Page 3 L6 "Riddick et al. (2016) treated a landfill site in Suffolk, UK as a point source and estimated a mean $CH_4$ emission of 709 $\mu g\ m^{-2}\ s^{-1}$ using $CH_4$ concentration data, collected 800 m from the landfill, and meteorological data in an inverse dispersion model. At a farther distance, 2 km, Hensen and Scharff (2001) used a Gaussian plume model to estimate emissions of between 66 and 292 $\mu g\ m^{-2}\ s^{-1}$ from three landfill sites near Amsterdam in the Netherlands. To our knowledge no research has been conducted on using a Gaussian plume approach at more than 2 km. Also, we believe that no other study has attempted to use an inversion model to identify emission hotspots within a |

| | | |
|---|---|---|
| | | landscape." |
| In section 2.3.3: Why was a Gaussian plume model used for the middle-distance analysis when NAME LPDM particle trajectories were available for Haddenham Church (and were used in the landscape analysis)? | The NAME model can calculate concentrations as well so I guess the two methods could have been used and cross-compared. However, the NAME trajectories would be more constrained because of the 3D met files used at 1.5km resolution and it was decided that a Gaussian Plume model would be better suited for our purposes. | |
| Is auto-correlation of error accounted for in the uncertainty analyses? Can the uncertainties be evaluated somehow (reduced chi-square statistic, cross-validation)? | Uncertainty correlation is not considered in InTEM, errors are considered independent of each other. This is one of InTEM's weaknesses and further analysis is needed using other cost functions can help with this (i.e. the Bayesian cost function). | |
| On page 5, line 4; the authors state that "Methane emissions are calculated using measured CH4 mixing ratio enhancement downwind, measured background CH4 mixing ratios upwind and the | As we only had access to one portable methane instrument a 15-minute averaged background methane concentration was measured upwind of the landfill site before, at 12 pm and after | Text added to Page 4 L20: "Background $CH_4$ mixing ratios were measured using the Los Gatos UGGA upwind of the landfill site before, at 12 pm and after each day's measurements and converted |

| | | |
|---|---|---|
| simulated ratio of CH4 mixing ratio enhancement to emission." There was no information given about the methodology used to measure and calculate the background mixing ratio. Please give the details on the methodology, show the data, and demonstrate that this is a valid representation of the background. Systematic errors in the background have a direct effect on the emissions estimate, so this is a critical aspect of the model. | each day's measurements. To show this text has been added to the manuscript. | into 15 minute averages." |
| Page 9, line 14: "A statistical filtering technique separated methane mixing ratios at each site into eight-time series...". Please give a description of the technique and a citation if applicable. Also, remove the hyphen on "eight-time". | The timeseries of each of the 4 site's concentrations is separated into 8 different timeseries (based on which 'octrant' the air history, calculated by NAME, has come from, e.g. NNW, NNE, SSE etc.). A rolling 18$^{th}$ percentile through these new 8 timeseries. See comment below for why. | A new paragraph is included (P9 L12) to make the link to the Connors et al paper, Sarah Connors' thesis and the new information in the supplementary material clearer. "The results presented here are taken from a study developing a method to estimate regional CH4 emissions in East Anglia |

| | | |
|---|---|---|
| | | (Connors et al., in prep.). More details on the measurements sites, the inversion set-up used for InTEM, the diagnostics used and the emission uncertainties can be found there and in Connors (2015). The main points for the purposes of this paper are summarised below and in the Supplementary Material." |
| Please provide the following inversion diagnostics to demonstrate that the assumptions implicit in the inversion framework are satisfied (These could be contained in supplementary material):

o Scatterplots of posterior enhancements vs. observed enhancements.

o Time-series of observations, background values, and posterior enhancements.

o Scatterplots of residuals vs. fitted values, demonstrating | Scatterplots of residuals vs. fitted values and the ime-series of observations, background values and posterior enhancements. have been added in Supplementary Material Section 2.

The scatterplots of residuals vs. fitted values and the time-series of residuals are in presented in Connors et al (in prep.) and the author would prefer to keep them in this publication alone.

The reduced chi-squared statistic is a Bayesian statistic and not available for the | |

| | | |
|---|---|---|
| homoscedasticity.

o Time-series of residuals, demonstrating that they do not drift

(especially for the landscape scale, where emissions are not temporally resolved, even though a seasonal cycle is demonstrated in the middle-distance analysis).

 o The reduced chi-squared statistic, demonstrating well-quantified uncertainties. | least-squares cost function. | |
| Page 1, line 17: "Landfill emissions have been estimated using three different approaches (WindTrax, Gaussian plume, and NAME InTEM inversion) applied to the measurements made close to the source and at Haddenham." This sentence is confusing because it does not accurately represent the scale of each method, and because it leaves out any | Corrected as suggested | Page 1 L17
"Landfill emissions have been estimated using three different approaches at different scales; near-source using the WindTrax inversion dispersion model, middle-distance using a Gaussian plume model and at the landscape scale using the NAME InTEM inversion." |

| | | |
|---|---|---|
| mention of the greater East Anglia Network. Please rephrase. | | |
| Page 1, line 21: "The estimated annual emissions vary between 11.6 and 13.7 Gg CH4 / yr." It is not initially clear to the reader if this is an uncertainty bounds or temporal variability. Please rephrase. | Corrected as suggested | Page 1 L22 changed text to: "with an estimated annual emission of 11.6 Gg $CH_4$ $yr^{-1}$" |
| Page 3, line 22: "This presents methane..." Should this be "this paper"? | Corrected as suggested | |
| Page 5, line 4: "Methane emissions are calculated using measured CH4 mixing ratio enhancement downwind, measured background CH4 mixing ratios upwind and..." please remove the first instance of the word "enhancement", as it is redundant in this context. | Corrected as suggested | |

Ms. Ref. No.: acp-2016-963

Title: Estimating the size of a methane emission point-source at different scales: from local to landscape

Department of Civil and Environmental Engineering
Princeton University
E320 Engineering Quad
Princeton
NJ

Email: sriddick@princeton.edu

23rd March 2017

Dear Editor,

We would like to thank the referee #2 for their comments. Please find our detailed responses below.

Yours sincerely,

Stuart Riddick (corresponding author)

and co-authors: Sarah Connors, Andrew Robinson, Alistair Manning, Pippa Jones, David Lowry, Euan Nisbet, Robert Skelton, Grant Allen, Joseph Pitt and Neil Harris

Reviewer #2

| My main problem in this paper is the conclusion that the landscape inverse modeling approach can be used to identify point sources. The inversion method lacks details and the discussion is somewhat superficial. I think OSSEs would be required to determine the ability of observations at the landscape scale to constrain emission hotspots. | We have refocused the paper and made the point that distinct emission sources can be observed within an emission landscape. We suggest that landscape inversion models can be used to identify emission hot-spots within an emission landscape. | Page 1 L25 the following has been removed: "is in good agreement with more labour-intensive near-source approaches and" |
|---|---|---|
| | | Page 1 L26 the following has been removed: "to provide high-quality emission estimates" |
| | | Page 12 L31 the following was removed: "agreement between the mid-distance estimates and the" and "that provide data for regional inversion models" |
| | | Page 13 L19 the following was added: "output from" |
| | | Page 12 L32 the following was removed: "the network and even to quantify their emissions hotspots" |
| | | Page 13 L20 the following |

| | | |
|---|---|---|
| | | was added "an emission landscape" |
| P6, L9-10: "The standard deviation of the lateral ($\sigma y$, m) and vertical ($\sigma z$, m) mixing ratio distribution are calculated from the stability class of the air (Pasquill, 1974)." So what are the values for the standard deviation used in this paper? | The values used can be found in the Supplementary Material Section 1. | |
| P7, L19: "This allows for any potential bias due to highly uncertain observations to be accounted for." I don't see how the bias would be accounted for. | High methane concentration values seen at Haddenham are usually short lived and appear as peaks lasting only a few hours (max). They usually occur at nighttime and, as the isotopic analysis shows, probably come from a landfill, which is an intermittent of methane. These are therefore more uncertain. The values would have a relatively high cost score at these times. So, by including the hourly SD into the uncertainty calculation this helps to de-weight the large concentrations, which | |

| | | |
|---|---|---|
| | have higher uncertainty, from increasing the overall cost score. | |
| P9, L14-15: "A statistical filtering technique separated methane mixing ratios at each site into. . ." What is this statistical filtering? | See comment above. | |
| P9, L16: Why "18th percentile"? Why not 10th or 25th? | This percentile is used as a result of sensitivity analysis showing that the resulting InTEM inversion results produced the lowest cost scores and therefore means the emissions produced are closer to the measured observations than any other percentiles tested. I tested from the 5$^{th}$ to the 45$^{th}$. Sensitivity analysis shows this baseline produces emission results with consistently stable emissions with the lowest cost score of all baselines tested. | |
| P9, L21: "For a more detailed description of the measurement sites and the | A new paragraph is included to make the link to the InTEM setup described in | Text added at P9 L12: "The results presented here are taken from a study |

| | | |
|---|---|---|
| InTEM setup please refer to Connors et al. (in prep)." I think more details about the InTEM setup should be given. For example, what prior constraints or regularization do you use? This is crucial for an inversion. | Connors et al. (in prep.), Sarah Connors' thesis and the new information in the supplementary material clearer.

This inversion does not use a prior, like the other studies referenced here. Priors are not essential but they are more commonly used than not. It uses a cost function similar to a least-squares approach. Bayesian cost-functions use priors and the analysis could (and probably should) be repeated with a Bayesian CF to comparison and a better assessment of errors. | developing a method to estimate regional CH4 emissions in East Anglia (Connors et al., in prep.). More details on the measurements sites, the inversion set-up used for InTEM, the diagnostics used and the emission uncertainties can be found there and in Connors (2015). The main points for the purposes of this paper are summarised below and in the Supplementary Material." |
| P12, L4: ". . .using near-source measurements are 453 kg hr-1 in June/July 2015. . ." I thought the near-source measurements cover only two days? This looks like two-month data. | Corrected as suggested | Added at p12 L16:
"30[th] June and 1[st] July 2015" |
| P12, L15-20: Table 4 shows the lowest emissions month is in April (111 kg/hr). I am not | This was typo and should be 1110 kg/hr and has been corrected.  The response of | |

| | | |
|---|---|---|
| very convinced that seasonality is due to temperature. Does stability class in the Gaussian plume approach play a role? | CH$_4$ emission from landfill to temperature is well documented and a result of methanotrophic bacteria becoming more active during the summer months. | |
| P12, L33-34: I am not convinced by this conclusion. See my general comments. | | Added at P12 L19:

[revised manuscript text omitted]

---

## Author Response (AR2)

Ms. Ref. No.:  acp-2016-963

Title: Estimating the size of a methane emission point-source at different scales: from local to landscape

Department of Civil and Environmental Engineering
Princeton University
E320 Engineering Quad
Princeton
NJ

Email: sriddick@princeton.edu

12$^{\text{th}}$ May 2017

Dear Editor,

We would like to thank the referee and editor for their comments. As suggested, we have amended the manuscript to incorporate all of the reviewers' comments.

Please find our detailed responses below.

Yours sincerely,

Stuart Riddick (corresponding author)

and co-authors: Sarah Connors, Andrew Robinson, Alistair Manning, Pippa Jones, David Lowry, Euan Nisbet, Robert Skelton, Grant Allen, Joseph Pitt and Neil Harris

Reviewer 1
Comment3

| In section 2.3.3: Why was a Gaussian plume model used for the middle-distance analysis when NAME LPDM particle trajectories were available for Haddenham Church (and were used in the landscape analysis)? | The NAME model can calculate concentrations as well so I guess the two methods could have been used and cross-compared. However, given the short distance from the landfill to the monitoring station and the availability of observed meteorology it was decided that the Gaussian Plume model would be better suited for our purposes. | The following was added at P6 L2: "The particle trajectories were calculated in the NAME model and could have been used the calculate emissions, however given the short distance from the landfill to the monitoring station and the availability of observed meteorology it was decided that a Gaussian Plume model would be better suited for our purposes." |
|---|---|---|

Comment 4

| Is auto-correlation of error accounted for in the uncertainty analyses? Can the uncertainties be evaluated somehow (reduced chi-square statistic, cross-validation)? | Uncertainty correlation is not considered in InTEM, errors are considered independent of each other. This is a weakness of the setup employed and further analysis is needed using other cost functions can help with this (i.e. the Bayesian cost function). | Added at P7 L30: "Uncertainty correlation was not considered in the modelling, errors are considered independent of each other. This is a weaknesses and further analysis is needed using other cost functions (e.g. the Bayesian cost function)." |
|---|---|---|

Comment P9 L14

| Have you removed the hyphen as requested for "eight-time series"? I agree with the reviewer, this should read "eight timeseries". | Changes as suggested at P9 L27. | P9 L27: "eight-time series" changed to "eight timeseries" |
|---|---|---|

Reviewer 2
Comment 2

| P6, L9-10: "The standard deviation of the lateral ($\sigma y$, m) and vertical ($\sigma z$, m) | The values used can be found in the Supplementary Material Section 1. | Text added at P6 L13: |
|---|---|---|

| mixing ratio distribution are calculated from the stability class of the air (Pasquill, 1974)." So what are the values for the standard deviation used in this paper? | | "the values used in our analyses are presented in Supplementary Material Section 1 (Pasquill, 1974)." |
|---|---|---|

Comment 3

| P7, L19: "This allows for any potential bias due to highly uncertain observations to be accounted for." I don't see how the bias would be accounted for. | High methane concentration values seen at Haddenham are usually short lived and appear as peaks lasting only a few hours (max). They usually occur at night time and, as the isotopic analysis shows, probably come from a landfill, which is an intermittent of methane. These are therefore more uncertain. The values would have a relatively high cost score at these times. So, by including the hourly SD into the uncertainty calculation this helps to de-weight the large concentrations, which have higher uncertainty, from increasing the overall cost score. | Text added at P7 L22: "High $CH_4$ concentration values seen at Haddenham are usually short lived and only appear as peaks lasting a maximum of only a few hours. These usually occur at night time and, as the isotopic analysis shows, probably come from a landfill, which is an intermittent of $CH_4$. These are therefore more uncertain. The values would have a relatively high cost score at these times. So, including an hourly standard deviation into the uncertainty calculation helps to de-weight the large concentrations, which have higher uncertainty, from increasing the overall cost score." |
|---|---|---|

Comment 4

| P9, L14-15: "A statistical filtering technique separated methane mixing ratios at each site into. . ." What is this statistical filtering? | See comment above. | A new paragraph is included (P9 L12) to make the link to the Connors et al paper, Sarah Connors' thesis and the new information in the supplementary material clearer. "The results presented here are taken from a study developing a method to estimate regional $CH_4$ |
|---|---|---|

| | | emissions in East Anglia (Connors et al., in prep.). More details on the measurements sites, the inversion set-up used for InTEM, the diagnostics used and the emission uncertainties can be found there and in Connors (2015). The main points for the purposes of this paper are summarised below and in the Supplementary Material Section 2." |
|---|---|---|

Comment 5

| P9, L16: Why "18th percentile"? Why not 10th or 25th? | This percentile is used as a result of sensitivity analysis showing that the resulting InTEM inversion results produced the lowest cost scores. Therefore, the emissions produced are closer to the measured observations than any of the other percentiles tested. We tested from the $5^{th}$ to the $45^{th}$. Sensitivity analysis shows this baseline produces emission results with consistently stable emissions with the lowest cost score of all baselines tested. | Text added at P9 L29: "This percentile was chosen as a result of a sensitivity analysis which showed that InTEM inversion results using the $18^{th}$ percentile produced the lowest cost scores, i.e. the calculated emissions are closer to the measured observations compared to any of the other percentiles tested (percentiles from the $5^{th}$ to the $45^{th}$ were tested)." |
|---|---|---|

| - Both reviewers suggest OSSEs to strengthen support for the landscape scale approach. Are the authors sure they do not wish to add this at this stage?

- Following on from this: In the current state, I think you need to remove the last sentence of the abstract. Both reviewers agree you cannot | Unfortunately, due to a lack of resources we are unable to perform OSSEs. As suggested the last sentence of the abstract has been removed. | |
|---|---|---|

| | | |
|---|---|---|
| "suggest the landscape inverse modelling approach described in this paper can be used to identify point-sources within an emission landscape" without additional support. | | |
| My issue is that the authors included a plot that shows troubling results. This is the plot under "Supplementary Material Section 2: Scatterplot of posterior enhancements vs. observed enhancements".

First: these are not enhancements, they are concentrations. In the inverse modeling literature an enhancement is the concentration of the constituent at the observation site *minus the background.* In my review I asked for a plot of modeled vs observed enhancements in order to separate the variability in the background from the variability in the influence of emissions. When one plots modeled vs. observed concentrations, a well modeled background will hide problems with the model of the emissions.

Second: the points in this plot are so dense that they are indistinguishable. There are plotting strategies that can ameliorate this. | The reviewer has raised an interesting point, but we disagree as to its importance. Before going into detail, we think there are two options: (a) remove the NAME InTEM material and put it in a later paper which describes the method in more detail; and (b) leaving the material in with more explanation and context as to the message we are trying to make with that material. We strongly prefer the latter as there is a real need to start addressing the issue of the consistency of GHG emission estimates across scales which is currently lacking.

The first point we would like to clarify is that the calculation in NAME InTEM does subtract out a baseline before the emissions are estimated. The procedure for baseline estimation is summarised in section 3.2.2 with more information given in Sarah Connors's thesis (now referenced) as well as in the Connors et al paper (in prep). In the inversion, NAME InTEM only calculates enhancements. The | Text added to manuscript before P13 L17, i.e. as a new penultimate paragraph: "Even though the annual emission estimate calculated using the InTEM inversion model is close to that calculated by the Gaussian Plume model, the uncertainty associated with the InTEM inversion estimate is large. Comparison of the measurements with the $CH_4$ time series produced by NAME InTEM (Supplementary Figures 1 and 2) shows the model to consistently underestimate the larger and sharper observed peaks. This arises as a result of the smaller weighting given to the peaks in the observed atmospheric concentrations in the NAME InTEM analysis (which uses all data) than in the WindTrax and Gaussian plume analyses which focus on these peaks. In particular, high peaks are underweighted because they are small scale features not easily delineated in the regional inversions and the boundary layer is harder to model accurately at night when the highest peaks tend |

Third, and **most importantly**, I think this plot indicates a problem in the model. There is a clear heteroscedasticity in the residuals. This is likely due to the fact that the background was not subtracted. The variation in the model and observed *background* is likely falling along the 1-1 line, and the enhancements are likely falling off of it, and are likely heteroscedastic.

A plot of modeled vs observed enhancements is generally considered the most basic diagnostic plot for this type of analysis. I would actually like to see this type of plot for every site and every model. It isn't clear for which model this plot is showing.

baseline values have been added back in to that figure.

The second point we are trying to make is that we are not trying to hide anything. While the data points are dense in that figure, they are much clearer in the accompanying time series showing the measurements and the calculated values. We have changed the order of those two figures in order to make the point more clearly. The figures could be redrawn, but it would take a few weeks.

Thirdly, we are not sure why this behaviour should be expected for a large-scale inversion model attempting to estimate point source emissions:

   (a)    The different models give different weightings to the large peaks observed in the concentrations. The emissions calculated by WindTrax and the Gaussian Plume model use those peaks as their major source of information. The NAME InTEM approach, on the other hand, tends to give them low statistical weighting because (i) it is hard to model such small-scale signals in the regional inversion and (ii) the

to occur due to their containment within the shallow nocturnal boundary layer. The heteroscedasticity seen in Supplementary Figure 2 is therefore to be expected as NAME InTEM reproduces the lower values better than the high ones.
The inherent challenges in inversion modelling, such as assuming a constant monthly emission (Supplementary Material Section 2 Figure SM2.3) and the atmospheric variability at night which is poorly resolved by the model, result in the emission estimates calculated in this research having an uncertainty of $\pm 91\%$.  This research is presented as an example of inversion modelling: a work in progress and, while the emission estimates are currently uncertain, the location of the emissions are well represented."

| | | |
|---|---|---|
| | events occur most strongly at night when the meteorological description is poorer.

(b)   There is a large variability in emissions (as shown in several Figures) while NAME InTEM is producing an annual estimate.  Further, the emissions are not normally distributed (see new Figure in Supplementary info).

The discrepancies shown in the figure are entirely consistent with these factors as the 'outliers' are nearly all occasions when the measurements are higher than the modelled values. The point of including the analysis in the paper (and it is not a major part) is to examine the consistency between the 3 approaches. We are not trying to exaggerate its importance, but we are trying to highlight its potential for (a) identification of point source emissions, and (b), in time, their quantification. On-going work is underway to improve baseline estimation and error analysis.

Finally, similar anomaly plots cannot be straightforwardly produced for the WindTrax and Gaussian Plume approaches because they solve for the emissions values which match the | |

| | observations. (In terms of the 3rd point above, they use all the information contained in each peak studied.) We are therefore unclear as to why putting these back into concentrations is meaningful. | |
|---|---|---|

[revised manuscript text omitted]